# Use of Three Air Samplers for the Detection of PRRSV-1 under Experimental and Field Conditions

**DOI:** 10.3390/ani8120233

**Published:** 2018-12-07

**Authors:** Heiko Stein, Jochen Schulz, Rebecca Morgenstern, Thomas Voglmayr, Georg Freymüller, Leonie Sinn, Till Rümenapf, Isabel Hennig-Pauka, Andrea Ladinig

**Affiliations:** 1Department for Farm Animals and Veterinary Public Health, University Clinic for Swine, University of Veterinary Medicine Vienna, 1210 Vienna, Austria; heiko.stein@googlemail.com; 2Institute for Animal Hygiene, Animal Welfare and Farm Animal Behaviour, University of Veterinary Medicine Hannover, Foundation, 30173 Hannover, Germany; jochen.schulz@tiho-hannover.de; 3Boehringer Ingelheim RCV GmbH & Co KG, 1120 Vienna, Austria; rebeccalanghoff@hotmail.com; 4Traunkreis Vet Clinic, 4551 Ried im Traunkreis, Austria; thomas.voglmayr@vetclinic.at (T.V.); georg.freymueller@vetclinic.at (G.F.); 5Institute of Virology, Department of Pathobiology, University of Veterinary Medicine Vienna, 1210 Vienna, Austria; leonie.sinn@gmx.de (L.S.); till.ruemenapf@vetmeduni.ac.at (T.R.); 6Field Station for Epidemiology, University of Veterinary Medicine Hannover, Foundation, 49456 Bakum, Germany; Isabel.hennig-pauka@tiho-hannover.de

**Keywords:** PRRSV, airborne, air sampler, aerosol

## Abstract

**Simple Summary:**

Respiratory diseases are of particular importance in pig production since they influence productivity, animal welfare and consumer protection. One of the most important respiratory diseases in pigs is caused by the porcine reproductive and respiratory syndrome virus (PRRSV). This virus, which can be divided into two separate species (PRRSV-1 and PRRSV-2), is causing high economic losses in the swine industry but does not represent a threat to humans, who cannot be infected with the virus. Airborne transmission of PRRSV has been reported to occur in PRRSV-2 isolates, but hardly any reports exist about airborne transmission of PRRSV-1. Therefore, we assessed three different air sampling systems for their ability to collect PRRSV-1 both under experimental conditions and in the field. First, we vaporized PRRSV-1 in an experimental chamber by a fogging device. Then, we tested the same air samplers under field conditions in a PRRSV-1 positive pig farm. Under experimental conditions all three air sampling systems were able to detect PRRSV-1. However, all three systems failed to detect PRRSV-1 under field conditions.

**Abstract:**

Airborne transmission of porcine reproductive and respiratory syndrome virus (PRRSV) has been known for a long time. Most experiments were performed using PRRSV-2 strains and fairly little information is available on the airborne spread of PRRSV-1. The aim of this study was to assess three different air samplers for their ability to detect PRRSV-1 under experimental and field conditions. All three devices were able to detect PRRSV-1 by quantitative reverse trascription polymerase chain reaction (qRT-PCR) under experimental conditions. However, the detection of PRRSV-1 in a PRRSV-positive farm with active virus circulation was not successful.

## 1. Introduction

Porcine reproductive and respiratory syndrome virus (PRRSV) is an economically important disease in pig production [1,2]. PRRSV is grouped in the genus Arterivirus and is an enveloped virus with single-stranded RNA genome of positive polarity [3]. PRRSV strains can be classified into two separate species: PRRSV-1 (former genotype 1 or European-type) and PRRSV-2 (former genotype 2 or North American) [4]. Airborne transmission of PRRSV has been known for a long time. Aerosol particles could carry infectious PRRSV [5]. Airborne transport of PRRSV has been described for distances up to 9.1 km [6]. Most studies investigating airborne transmission of PRRSV were performed with PRRSV-2 strains. Hardly any research in peer-reviewed journals has reported airborne transmission of PRRSV-1 strains. The aim of this study was to assess three different air samplers for their ability to detect aerosolized PRRSV-1 under experimental conditions. The same air samplers were tested under field conditions in one nursery and two fattening units of a PRRSV positive farm with active virus circulation.

## 2. Materials and Methods

### 2.1. Sample Collection in the Experimental Chamber

In an isolation unit with an air volume of 68.5 m^3^, one liter of Phosphate Buffered Saline (PBS) containing 1 × 10^7^ TCID_50_ (median tissue culture infective dose, equaling 2.6 × 10^11^ virus copies by qRT-PCR) of a PRRSV-1 modified live vaccine (ReproCyc^®^ PRRS EU, Boehringer Ingelheim Vetmedica GmbH, Ingelheim/Rhein, Germany) was nebulized for 10 min with a cold-fogging system (UNIPRO2, IGEBA, Weitennau, Germany). Three air samplers were tested: Coriolis^®^μ (Bertin Technologies, Montigny-le-Bretonneux, France, flow rate 3 m^3^/10 min), MD8 Airscan (Sartorius Stedim Biotech GmbH, Göttingen, Germany, flow rate 1 m^3^/10 min) and two IOM Multidust samplers (SKC Inc., Eighty Four, PA, USA) with polycarbonate filters (pore size of 0.2 μm, flow rate 2.5 L/min). Two air collections of ten minutes were performed with Coriolis^®^μ and MD8 Airscan. The first sampling was carried out during nebulisation and the second sampling one hour later. The air sampling with two IOM Multidust samplers started simultaneously with nebulisation and continued for a period of two hours. For Coriolis^®^μ, 15 mL cell culture medium (MEM + 3% FCS, Minimum Essential Medium and Fetal Calf Serum) was used, the sampled air was conducted through the cell culture medium. MD8 Airscan sampled air through a Gelatine filter which was dissolved in 20 mL MEM by heating to 37 °C. The air of the IOM Multidust sampler passed through the polycarbonate filters which were washed in 10 mL MEM. Particles ≥ 200 nm were mostly captured by the 200 nm rated Nuclepore filter [7]. Results obtained by qRT-PCR (RNA copies/mL) were used to calculate RNA copies per cubic meter air after accounting for the total volume of MEM used and air volume collected by each air sampler. After sample collection, MEM was tested for the presence of PRRSV by qRT-PCR and virus isolation was performed on MARC-145 cells (CCLV RIE 277) as described elsewhere [8]. 

### 2.2. Sample Collection in the Field

Air samples were collected in one nursery and two fattening units (fattening unit 1 and 2) in a commercial farm positive for PRRSV-1. Air samples were collected in all units for ten minutes with Coriolis^®^μ and MD8 Airscan. Measurements with IOM Multidust samplers were performed over a period of 112 min in the nursery unit and over 102 min in fattening unit 1. Serum and saliva samples were taken by the herd veterinarian for routine monitoring of the PRRSV status of the herd and confirmed that the pigs were positive for PRRSV-1. Since samples from pigs were collected for routine diagnostics, no animal ethics approval was required. Two cotton ropes were used to collect oral fluid samples in each unit and serum samples were collected from 15 pigs in each unit. Five serum samples were pooled to one sample. Saliva, serum and air samples were tested for the presence of PRRSV by qRT-PCR and virus isolation on MARC-145 cells. While pooled serum samples were tested by qRT-PCR, individual serum samples were used for virus isolation. 

## 3. Results

### 3.1. Experimental Chamber

Results for the detection of PRRSV-1 in the medium of the three different air samplers are shown in Table 1. The highest RNA concentrations were measured by qRT-PCR in samples collected with Coriolis^®^μ during nebulisation (9 × 10^8^ RNA copies/m^3^ air) and one hour later (2.4 × 10^8^ RNA copies/m^3^ air). In MD8 Airscan samples 3.0 × 10^8^ and 1.1 × 10^8^ RNA copies/m^3^ air were measured during and after nebulisation. In samples of the two IOM Multidust samplers, RNA concentrations were 2.1 × 10^7^ and 1.7 × 10^7^ copies/m^3^ air. The control samples and transport control samples were negative by qRT-PCR. PRRSV could only be isolated from the sample collected during nebulisation with Coriolis^®^μ at a concentration of 1.5 × 10 TCID_50_/mL. 

### 3.2. Field Samples

Results are summarized in Table 2. None of the three air samplers was able to collect PRRSV in amounts detectable by qRT-PCR. All oral fluid samples were positive by qRT-PCR except for samples from fattening unit 1. Pooled serum samples were PRRSV positive by qRT-PCR except for one serum-pool from fattening unit 1. The virus could be isolated from at least one serum sample out of the serum sample pools which were positive by qRT-PCR (Table 2). 

## 4. Discussion

All studies describing PRRSV detection in air samples in peer-reviewed journals used PRRSV-2 isolates. However, there is one non-peer-reviewed report describing detection of PRRSV-1 in air samples collected inside a PRRSV positive farm and 30 m downwind from four PRRSV-positive herds. The detection of PRRSV inside barns was performed during mass vaccination, which might have led to massive spread of the virus, allowing the detection of PRRSV-1 in air samples. No detailed description on the technical data of the air sampler system was given and therefore results cannot be compared with our study [9]. The results of our study are similar to those obtained by Trincado et al. who aerosolized a PRRSV-2 strain in an experimental aluminum chamber and took air samples with an all-glass impinger [10]. They could detect PRRSV-2 in air samples by PCR. Based on our results obtained under experimental settings, Coriolis^®^μ was the most sensitive of the air samplers tested in the present study. In a subsequent experiment, Trincado et al. placed two PRRSV-2 infected pigs in an experimental chamber. While serum and swabs of those pigs were positive for PRRSV by PCR, air samples were negative. The authors assume that two pigs do not generate a sufficient amount of infectious aerosol particles to be detected by PCR. The authors also speculate that the virus isolate might not have replicated to amounts detectable by the experimental setup [10]. Dee et al. also noted that a certain number of pigs is needed to generate infectious aerosols [11]. In a different study, 116 pigs were infected with PRRSV-2 and subsequently ten sentinel pigs were exposed to air exhausted from the barn of the infected pigs. The sentinel pigs did not become infected and all air samples were negative for PRRSV [12]. If the pathogenicity of the particular PRRSV isolate plays a role in the airborne transmission of the virus was tested in another study. Pigs infected with a low or a highly pathogenic strain of PRRSV-2 were used to measure infection of sentinel pigs, which were exposed to aerosols from the infected pigs. Only the RNA of the highly pathogenic isolate was detectable in air samples and four of ten sentinel pigs exposed to aerosols containing the highly pathogenic isolate became infected [13]. An older study of Cho et al. showed that the likelihood of aerosol shedding of a highly pathogenic PRRSV-2 isolate was significantly increased compared to a low pathogenic strain [14]. An inherent problem of defining the virus load in aerosols is the sensitivity of PRRSV detection. Infection experiments with sentinel pigs are by far the most sensitive assays, as: (1) pigs are exposed over a longer period of time in which they are able to exchange large volumes of air; (2) the virus is directly guided to susceptible tissues; and (3) a productive infection is initiated by inhaling a few infectious particles. In contrast, the lower reliable detection limit of viral genomes by RT-PCR is about 10–100 genome equivalents per reaction. Due to the methods of nucleic acid extraction and RT-PCR dilution factors of >100-fold minimal PRRSV concentrations of 10^4^–10^5^/mL collection fluid will be required for detection. PRRSV isolation in cell culture is not very sensitive (depending on virus isolate and cell type) and prone to contamination. Also, the detection of virus particles under field conditions is not comparable to an artificially produced particle. Virus particles in pig barns are exposed to various environmental factors which may damage their structure. Furthermore, it is possible to speculate that the sampling system itself harms the virus particles. All three systems were able to collect PRRSV-1 in amounts measurable by qRT-PCR. The environmental conditions in our field settings, the unknown number of virus shedding pigs leading to unknown amounts of aerosolized PRRSV, negative influences in aerial environment in the field, as well as the specific PRRSV isolate present in the farm might have resulted in negative air samples collected in the field. As confirmed by the analyses of serum samples and oral fluid samples, pigs on the investigated farm seemed to have relatively low levels of PRRSV replication. Therefore, it can be speculated that they also shed low levels of aerosolized virus particles to be picked up by the air sampling systems. Further studies are necessary to investigate air sample collection under field conditions and to optimize sample collection and processing in order to evaluate the role of aerosols in transmission of PRRSV-1. 

## 5. Conclusions

Under experimental conditions, Coriolis^®^μ, MD8 Airscan and IOM Multidust samplers with polycarbonate filters were able to detect aerosolized PRRSV-1. Of the three air samplers tested, Coriolis^®^μ was the most sensitive in detecting aerosolized PRRSV. In contrast, all three air sampling systems failed to detect PRRSV-1 in one PRRSV-positive farm, which might be due to several reasons including low levels of virus shedding by infected pigs, the specific PRRSV field isolate present on the farm, environmental factors influencing virus particles, the detection limit of air samplers and also laboratory methods, and so on. Therefore, further studies are needed to assess the role of aerosols in PRRSV-1 transmission.

## Figures and Tables

**Table 1 animals-08-00233-t001:** Porcine reproductive and respiratory syndrome virus (PRRSV)-1 detection in air samples collected in experimental chamber.

Air Sampler	Time Point	qRT-PCR Copies/mL (Ct-Values)	Virus Isolation
t-valueCoriolis^®^μ	During nebulisation	1.8 × 10^8^ (20.74)	1.5 × 10 TCID_50_
	1 h after nebulisation	4.7 × 10^7^ (22.82)	negative
MD8 Airscan	During nebulisation	1.5 × 10^7^ (24.54)	negative
	1 h after nebulisation *	5.5 × 10^6^ (26.07)	negative
IOM Multidust sampler 1	2 h after start of nebulisation	6.3 × 10^5^ (29.37)	negative
IOM Multidust sampler 2	2 h after start of nebulisation	5.2 × 10^5^ (29.64)	negative

Results of qRT-PCR and virus isolation performed on air samples collected by the three air sampling systems under experimental conditions in an experimental chamber. * Sampling time was seven minutes due to a switch-off of the sampler. TCID: tissue culture infective dose. Ct-value: threshold cycle value.

**Table 2 animals-08-00233-t002:** PRRSV-1 detection in air samples collected in the field.

Samples/Unit	qRT-PCR Copies/mL (Ct-Values)	Virus Isolation
Air Coriolis^®^μ nursery unit	negative	negative
Air MD8 Airscan nursery unit	negative	negative
Air IOM Multidust nursery	negative	negative
Oral fluid #1 nursery unit	3.2 × 10^4^ (33.83)	negative
Oral fluid #2 nursery unit	1.0 × 10^4^ (35.58)	negative
Serum pool 1–5 nursery	6.8 × 10^3^ (36.21)	1 positive sample (#5)
Serum pool 6–10 nursery	1.7 × 10^5^ (31.37)	2 positive samples (#7, #8)
Serum pool 11–15 nursery	6.4 × 10^6^ (25.86)	2 positive samples (#13, #14)
Air Coriolis^®^μ fattening unit 1	negative	negative
Air MD8 Airscan fattening unit 1	negative	negative
Air IOM Multidust fattening unit 1	negative	negative
Oral fluid #1 fattening unit 1	negative	negative
Oral fluid #2 fattening unit 1	negative	negative
Serum pool 16–20 fattening unit 1	1.1 × 10^4^ (35.55)	1 positive sample (#19)
Serum pool 21–25 fattening unit 1	negative	negative
Serum pool 26–30 fattening unit 1	1.8 × 10^3^ (38.17)	1 positive sample (#27)
Air Coriolis^®^μ fattening unit 2	negative	negative
Air MD8 Airscan fattening unit 2	negative	negative
Oral fluid #1 fattening unit 2	2.5 × 10^4^ (34.22)	negative
Oral fluid #2 fattening unit 2	1.1 × 10^4^ (35.55)	negative
Serum pool 31–35 fattening unit 2	1.4 × 10^5^ (31.68)	1 positive sample (#35)
Serum pool 36–40 fattening unit 2	4.9 × 10^4^ (33.21)	2 positive samples (#36, #40)
Serum pool 41–45 fattening unit 2	9.0 × 10^4^ (32.29)	5 positive samples

Results of qRT-PCR and virus isolation performed on air samples collected by the three air samplers as well as in oral fluid and serum samples collected in a PRRSV-1 positive nursery and two fattening units.

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
