# Peer review of "Use of Three Air Samplers for the Detection of PRRSV-1 under Experimental and Field Conditions"

_animals, 2018, doi:10.3390/ani8120233_

Round 1
Reviewer 1 Report
L51- it should say: "has been known"
Could you clarify in the experimental part of the study, how many samples were taken with each sampler? Also, did the authors test the inoculum being used for nebulization before and after it was done (did they take an aliquout in particular at the end of the nebulization of what was left?). Please report if so. This is useful to interpret the data..
Confirm that air samples were tested individually for PCR and not pooled
Please add also the cycle threshold values of the RT-PCR samples.
Based on the data presented it seems that there was very limited amount of PRRS virus in the populations sampled. It's hard to make conclusions of the performance of the samplers when the animals don't seem to have shed much. The discussion does not address this point fully. Most of the discussion is about differences in strains ability to be aerosolized, which has been reported, but it does not discuss in detail the fact that there didn't seem to be much virus in the population in the first place. This can be seen based on the results from the PCR copies/blood or oral fluids. If samples are still available, authors could consider testing the samples of the pigs and oral fluids individually rather than pools in order to better discuss why the samplers were not able to perform well under field conditions. As the study is presented, there is some confounding of factors that should be addressed. Authors should also discuss differences in sensitivity of the samplers based on their experimental data.
Author Response
L51- it should say: "has been known".
Response: Wording has been changed
Confirm that air samples were tested individually for PCR and not pooled.
Response: All air samples have been tested individually. The only samples which have been pooled were the serum samples from pigs on the farm.
Please add also the cycle threshold values of the RT-PCR samples.
Response: CT-values have been added to the tables.
Based on the data presented it seems that there was very limited amount of PRRS virus in the populations sampled. It's hard to make conclusions of the performance of the samplers when the animals don't seem to have shed much. The discussion does not address this point fully. Most of the discussion is about differences in strains ability to be aerosolized, which has been reported, but it does not discuss in detail the fact that there didn't seem to be much virus in the population in the first place. This can be seen based on the results from the PCR copies/blood or oral fluids. If samples are still available, authors could consider testing the samples of the pigs and oral fluids individually rather than pools in order to better discuss why the samplers were not able to perform well under field conditions. As the study is presented, there is some confounding of factors that should be addressed. Authors should also discuss differences in sensitivity of the samplers based on their experimental data.
Response: We appreciate this comment by the reviewer and added two sentences to the discussion (lines 169-172). All oral fluid samples, which already represent samples on group level, were analysed individually, not in pools. Only serum samples were pooled, since according to our experience the pooling of up to five serum samples provides very reliable results.
Another sentence was added to the discussion (lines 133-135) and also in the conclusions (177-183) on the sensitivity of tested air samplers as suggested.
Reviewer 2 Report
The authors present the results of a study aimed at the comparison of three different air samplers for their ability to detect PRRSV-1 under experimental and field conditions.
They used bimolecular assays and virus isolation as measurement of PRRSV-1 presence (and quantification)
The study design is appropriate and poses the basis for future necessary further studies.
The text is well written and the reported results discussed extensively with respect to the (scarce) existing literature.
I have some (minor) comments to be addressed (if the authors consider them appropriate they could be added to manuscript).
1) the authors measured PRRSV-1 by qRT-PCR in the medium and then derived the concentration of the RNA copies /cubic meter . It's possible to consider the Coriolis system as more sensitive with respect to the others two samplers? or at least in comparison with the MD8 Airscan (i.e. if homogeneous "nebulization" and same sampling scheme, the result should speak in favor of different ability in "capturing" PRRSV...)
2) Do the authors have any information regarding the field strain circulating in the positive farm? (biological behavior, sequence...) this could help some of the speculations reported in the discussion...
3) In the discussion authors refer to the possible unknown amount of aerosolized PRRSV in the field. This aspect (i.e. the measurement of it) will not be solved with further field studies (because of the intrinsic variability of shedding pigs in a positive population) if not firstly assessing a sort of detection limit of the samplers system under controlled (experimental) conditions
4) The discussion is mostly focused on what can determine an "infectious aerosol" in field condition and which could be the factors influencing the sensitivity of virus isolation.
I'm consequently wondering if the present title is the best one for this paper, since the authors do not effectively compare the three air samplers; they just use three different air samplers, report the results of PRRSV detection but any discussion regarding these differences are reported. I Would skip the word "Comparison" from the title and rephrase it accordingly.
5) very few typos are present in the text:
abstract: qRT-PCR is define quantitative realtime polymerase chain reaction
RT refers to Reverse Trascription and not to RealTime
line 61 and 66: m3 --> write using apex
line105: switch- off --> switch-off
Author Response
Response to comments of Reviewer #2:
The authors present the results of a study aimed at the comparison of three different air samplers for their ability to detect PRRSV-1 under experimental and field conditions.
They used bimolecular assays and virus isolation as measurement of PRRSV-1 presence (and quantification)
The study design is appropriate and poses the basis for future necessary further studies.
The text is well written and the reported results discussed extensively with respect to the (scarce) existing literature.
I have some (minor) comments to be addressed (if the authors consider them appropriate they could be added to manuscript).
1) the authors measured PRRSV-1 by qRT-PCR in the medium and then derived the concentration of the RNA copies /cubic meter . It's possible to consider the Coriolis system as more sensitive with respect to the others two samplers? or at least in comparison with the MD8 Airscan (i.e. if homogeneous "nebulization" and same sampling scheme, the result should speak in favor of different ability in "capturing" PRRSV...)
Response: Indeed, since the volume of collect air by each sampler was accounted for, the Coriolis system was the most sensitive under our experimental conditions. We added a sentence to the discussion (lines 133-135) and also in the conclusions (177-183).
2) Do the authors have any information regarding the field strain circulating in the positive farm? (biological behavior, sequence...) this could help some of the speculations reported in the discussion...
Response: Unfortunately, we do not have any information on the sequence of the PRRSV isolate on the investigated farm. Attempts to sequence ORF5 of the isolate failed in the collected samples.
3) In the discussion authors refer to the possible unknown amount of aerosolized PRRSV in the field. This aspect (i.e. the measurement of it) will not be solved with further field studies (because of the intrinsic variability of shedding pigs in a positive population) if not firstly assessing a sort of detection limit of the samplers system under controlled (experimental) conditions
Response: We completely agree with the reviewer. We have also used serial dilutions of PRRS vaccine virus to be aerosolized under our experimental settings. We have not included those results due to the limited number of words available for short communications in Animalsand also due to the fact that the results cannot be transferred to the field situation, since many factors are going to influence results in the field (e.g. the particular PRRSV isolate, number of pigs infected and shedding, air volume, etc.).
4) The discussion is mostly focused on what can determine an "infectious aerosol" in field condition and which could be the factors influencing the sensitivity of virus isolation.
I'm consequently wondering if the present title is the best one for this paper, since the authors do not effectively compare the three air samplers; they just use three different air samplers, report the results of PRRSV detection but any discussion regarding these differences are reported. I Would skip the word "Comparison" from the title and rephrase it accordingly.
Response: We agree with the reviewer and have changed the title in order to avoid the word “comparison”.
5) very few typos are present in the text:
abstract: qRT-PCR is define quantitative realtime polymerase chain reaction
RT refers to Reverse Trascription and not to RealTime
line 61 and 66: m3 --> write using apex
line105: switch- off --> switch-off
Response: thank you for identifying those typos; we have corrected them.
Round 2
Reviewer 1 Report
It's good that the authors changed the title to say "use" instead of comparison. Similarly, the objective of the study should be updated to reflect the true scope of the project. L40 instead of "compare" could say "assess".
L41 - use lower case to spell out reverse transcription
When referring to the virus, use lower case as follows: porcine reproductive and respiratory syndrome virus
L55- change compare with "assess"
L61 - should say "liter" instead of "litre"
L77- should say "meter" instead of "metre"
L88- should say "diagnostics"
Author Response
Response to comments of Reviewer #1:
It's good that the authors changed the title to say "use" instead of comparison. Similarly, the objective of the study should be updated to reflect the true scope of the project. L40 instead of "compare" could say "assess".
Response: We have used “assess” instead of “compare” throughout the manuscript; this is true for lines 40, 55, but also 31
L41 - use lower case to spell out reverse transcription
Response: Changes have been made
When referring to the virus, use lower case as follows: porcine reproductive and respiratory syndrome virus
Response: Lower cases are used now
L61 - should say "liter" instead of "litre"
Response: Changes were made accordingly
L77- should say "meter" instead of "metre"
Response: Changes were made accordingly
L88- should say "diagnostics"
Response: Thanks, the typo has been changed
Reviewer 2 Report
The authors improved in a satisfactory way the manuscript according to the suggestions and substantially answer to all the questions raised.
Round 3
Reviewer 1 Report
The revised version looks good, I don't have any further comments.